# A standardized database of Last interglacial (MIS 5e) sea-level indicators in Southeast Asia

Kathrine Maxwell[1,2], Hildegard Westphal[1,2], Alessio Rovere[2,3]

[1]Leibniz Centre for Tropical Marine Research (ZMT), Bremen, Germany

[2]University of Bremen, Department of Geosciences, Germany

[3]MARUM - Center for Marine Environmental Sciences, University of Bremen, Bremen, Germany

*Correspondence to*: Kathrine Maxwell (kathrine.maxwell@leibniz-zmt.de)

**Abstract.**

Marine Isotope Stage 5e (the Last Interglacial, MIS 5e, 125 ka) represents a process analogue for a warmer world. Analysis of sea level proxies formed in this period helps in constraining both regional and global drivers of sea-level change. In Southeast Asia, several studies have reported elevation and age information on MIS 5e sea-level proxies, such as fossil coral reef terraces or tidal notches, but a standardized database of such data was hitherto missing. In this paper, we produced such sea-level database using the framework of the World Atlas of Last Interglacial Shorelines (WALIS, https://warmcoasts.eu/world-

atlas.html). Overall, we screened and reviewed 14 studies on Last Interglacial sea-level indicators in Southeast Asia, from which we report 43 proxies (42 coral reef terraces and one tidal notch), that were correlated to 134 dated samples. Five data points date to MIS 5a (80 ka), six data points are MIS 5c (100 ka), and the rest are dated to MIS 5e. The database compiled in this study is available at https://doi.org/10.5281/zenodo.4681325 (Maxwell et al., 2021).

## 1 Introduction

The Last Interglacial (LIG, about 129–116 ka), also referred to as Marine Isotope Stage 5e (MIS 5e), is often considered as a process analogue for predicted future changes for a warmer world (Burke et al., 2018). The LIG was characterized by global mean annual temperatures of 0.8 °C (maximum 1.3 °C) higher than pre-industrial ones (Fischer et al., 2018). $CO_2$ concentration was 250–285 parts per million by volume (ppmv) (Petit et al., 1999; Rundgren and Bennike, 2002) and ice sheets were smaller than today (i.e., Greenland and possibly Antarctic) (NEEM community members, 2013; Turney et al., 2020). As a consequence,

LIG sea level was higher than present. The LIG is one of the most studied time intervals for sea-level reconstructions, that are mostly based on relative sea level (RSL) proxies (such as marine terraces, coral reef terraces, and marine notches) preserved at several locations around the world (Pedoja et al., 2011; Hibbert et al., 2016). Proxy-based estimates coupled with spatio-temporal statistical and GIA modeling suggest that global mean sea level (GMSL) during the peak of the LIG was 6-9 m higher than present (Kopp et al., 2009; 2013).

In Southeast Asia, several studies have undertaken mapping and dating of LIG RSL proxies, but a standardized database similar to the one compiled with Holocene sea-level data (Mann et al., 2019) was hitherto not available. In this paper, we built such database using the framework of the World Atlas of Last Interglacial Shorelines (WALIS, https://warmcoasts.eu/world-atlas.html). We screened a total of 14 published studies addressing geological descriptions of LIG sea-level indicators. From these, we identified 43 unique RSL proxies, that were correlated with 134 dated samples (one RSL indicator may be correlated to many dated samples). Among these 134 samples, 21 were selected from the WALIS U-series database by Chutcharavan and Dutton (2020), in order to avoid duplications of samples within WALIS. Despite our work mostly being aimed at compiling MIS 5e data, we also inserted in our database MIS 5a and MIS 5c sites whenever they were not already in the database at the time of compilation (e.g., Thompson and Creveling, 2021). This paper serves the scope of providing accessory information on the compiled data. With this effort under the WALIS framework, gaps in the LIG literature in SE Asia are identified and potential areas for future research are pointed out.

## 2 Literature Overview

### 2.1 Tectonic setting of Southeast Asia

Southeast Asia is a tectonically complex region, characterized by the interaction of three major plates: the Eurasian Plate, the Indian–Australian Plate, and the Pacific–Philippine Sea Plate (Fig. 1). Eastern Indonesia (including Sulawesi and the islands of Sumba-Timor-Alor) is situated at the junction of these three major plates. Hall and Wilson (2000) describe five suture zones in eastern Indonesia: Molucca, Sorong, Borneo, Sulawesi, and Banda. The areas where fossil coral reef terraces are documented lie on the Sulawesi and Banda sutures. The Sulawesi Suture is the most complex one in the region, and is characterized by Cenozoic collision between continental, ophiolitic, and island arc fragments (Hall and Wilson, 2000). In Sulawesi, continent–continent collision and uplift began in the early Miocene, related to west-dipping subduction and collision with continental fragments derived from the Australian margin. Active deformation continues to the present day and a complex pattern of block rotations, strike–slip faulting linked to subduction at presently active trenches, is revealed by GPS measurements and geological mapping (e.g., Hall and Wilson, 2000). In the Banda Suture, very early stages of the collision of a continental margin with an island arc can be observed (e.g., Hall and Wilson, 2000; Harris, 2011). The Banda arc continent collision is characterized by a complex array of island arcs, marginal basins, continental fragments, and ophiolites that are amalgamated by repeated plate boundary reorganizations over the past 200 Ma (e.g., Harris, 2011). The collision of the Banda Arc, which consists of an inner volcanic arc and an outer non-volcanic arc of islands, with the Australian continental margin proves to be one of the best examples of active collision in the world since deformation is continuing to the present day (e.g., Hall and Wilson, 2000).

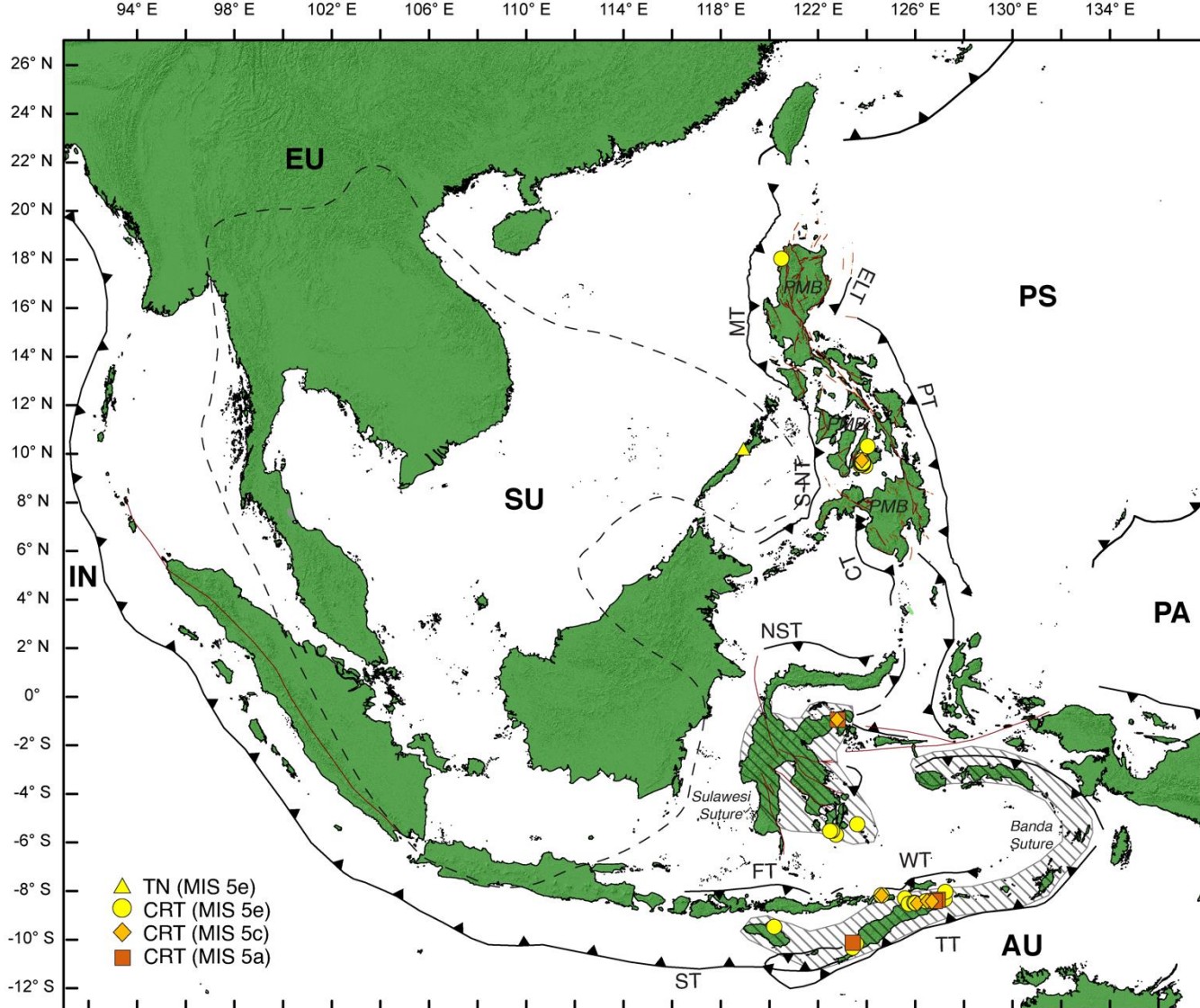

**Figure 1. Regional tectonic map of Southeast Asia and the locations of the 43 RSL indicators screened in this study. The region is being deformed by the movements of the Eurasian Plate (EU), the Indian (IN)–Australian (AU) Plate, and the Pacific (PA)– Philippine Sea (PS) Plate. Also shown is the extent of the Sundaland (SU) block from Simons et al. (2007). The Philippine archipelago is situated between the PS Plate to the east and the EU Plate to the west and the oblique convergence of these plates results in the formation of the Philippine Mobile Belt (PMB). The Philippines is bounded by oppositely-dipping subduction zone systems: Manila Trench (MT), Sulu-Negros Trench (S-NT), Cotabato Trench (CT), East Luzon Trough (ELT), and Philippine Trench (PT). Sulawesi is bounded to the north by the North Sulawesi Trench (NST) and lies on the Sulawesi Suture (Hall and Wilson, 2000). The Sumba-Timor-Alor region, situated between the EU Plate and the IN-AU Plate, lies on the Banda Suture (Hall and Wilson, 2000) and is bounded to the north by the Flores Thrust (FT) and Wetar Thrust (WT) and by the Sunda Trench (ST) and Timor Trough (TT) to the south. The MIS 5e RSL indicators (coral reef terraces (CRT) and tidal notch (TN)) reviewed in this study are distributed in these three regions. Also shown are the locations of MIS 5a and 5c CRT. Active faults (red lines) in the Philippines are from PHIVOLCS (2016) and active faults in Sulawesi are from Maulana et al. (2019).**

The Philippine archipelago is presently situated between two converging major plates: the Philippine Sea Plate to the east and the Eurasian Plate to the west. Oblique convergence between these two plates has resulted in the formation of the Philippine Mobile Belt (PMB), a 400-km-wide deformation zone from Luzon to Mindanao (Philippines), that is bounded by oppositely-dipping subduction zones (the discontinuous Manila- Sulu- Negros-Cotabato Trench system to the west and the East Luzon Trough-Philippine Trench system to the east) (Gervasio, 1966; Rangin et al., 1999). Accommodating the lateral component of this convergence is the 1200-km-long left-lateral strike-slip Philippine Fault that traverses the PMB from northwest Luzon to eastern Mindanao (Fitch, 1972; Barrier et al., 1991; Aurelio, 2000). The Cretaceous-to-Present geologic history of the Philippines has been shaped by various tectonic processes such as continent–arc collision, continental rifting, oceanic spreading, and multiple episodes of subduction (Aurelio et al., 2013). At present, the deformation in the Philippines, as revealed through the dense seismicity data since the 1900s (Philippine Institute of Volcanology and Seismology - PHIVOLCS, 2016), is predominantly controlled by movement along tectonic structures such as subduction zones and numerous onshore and offshore faults.

While the eastern portion of Southeast Asia is situated at the junction of major tectonic plates, the western portion of the region lies inside the Sundaland block. The core of Sundaland block encompasses mainland Southeast Asia (Cambodia, Laos, Vietnam, Thailand), the Malay Peninsula, Borneo, Sumatra, Java, and the Sunda shelf and has low rate of shallow seismicity (Simons et al., 2007). Recent work, which combined geomorphological observations with numerical simulations of coral reef growth and shallow seismic stratigraphy, suggests that the Sundaland is subsiding with transient dynamic topography the likely cause of subsidence (Sarr et al., 2019 and references therein).

The complex tectonic setting of Southeast Asia is reflected in the varied elevations of MIS 5e shorelines found at different locations. As an example, the highest shoreline is found in Alor Island, Indonesia, and has been uplifted at nearly 150 m above present sea level by neotectonics movements.

## 2.2 Data screening

The database presented in this study has been compiled with data from RSL indicators originally reported by previous works, that were inserted into the WALIS database if they met all of the following screening conditions.

1. **Geographic location.** Datapoints were inserted only if geographic coordinates for the sites/samples were provided or could be estimated from maps or indications contained in the original papers. When coordinates were not provided in the original publication, we used Google Earth to estimate the locations from the publications' maps.

2. **Elevation.** We included RSL indicators for which elevations were provided, with reference to modern sea level or a sea level datum. In most cases, measurement techniques (i.e., barometric altimeter, differential GPS, theodolite and rod, total station or auto/hand level) and elevation uncertainties are reported. In cases where the elevation measurement technique or the sea level datum was not reported, we followed the procedures described in the section on elevation measurement (Section 4).

3. **Ages.** RSL indicators were included in this database only if their ages were constrained by U-series or by Electron Spin Resonance (ESR) dating of corals or mollusks. In most of the studies, more than one sample is dated from one terrace tract. In these cases, all published coral dates (whether they are reliable or not) have been inserted into the database, and the assigned age (MIS 5e, 5c, or 5a) of that RSL indicator is based on the original interpretation of the published studies. Unreliable ages were identified and details about the dated samples (e.g., taxonomic information, in-situ or not, calcite content, analytical details) were also inserted in the database when available. A separate section describes the dating techniques we encountered in our review (Section 5).

## 3. Sea-level indicators

Most of the studies reviewed for this database are based on fossil coral reef terraces as sea-level indicators. In one site in the Philippines, a tidal notch used as paleo sea-level marker is reported (Omura et al., 2004). In this section we will describe the RSL indicators examined and their relationship with sea level (Fig. 2). For each data point inserted in our database, we assigned a quality score ranging from 1 (Poor) to 5 (Excellent), following the WALIS guidelines (Rovere et al., 2020), available at: https://walis-help.readthedocs.io/en/latest/RSL_data.html#quality.

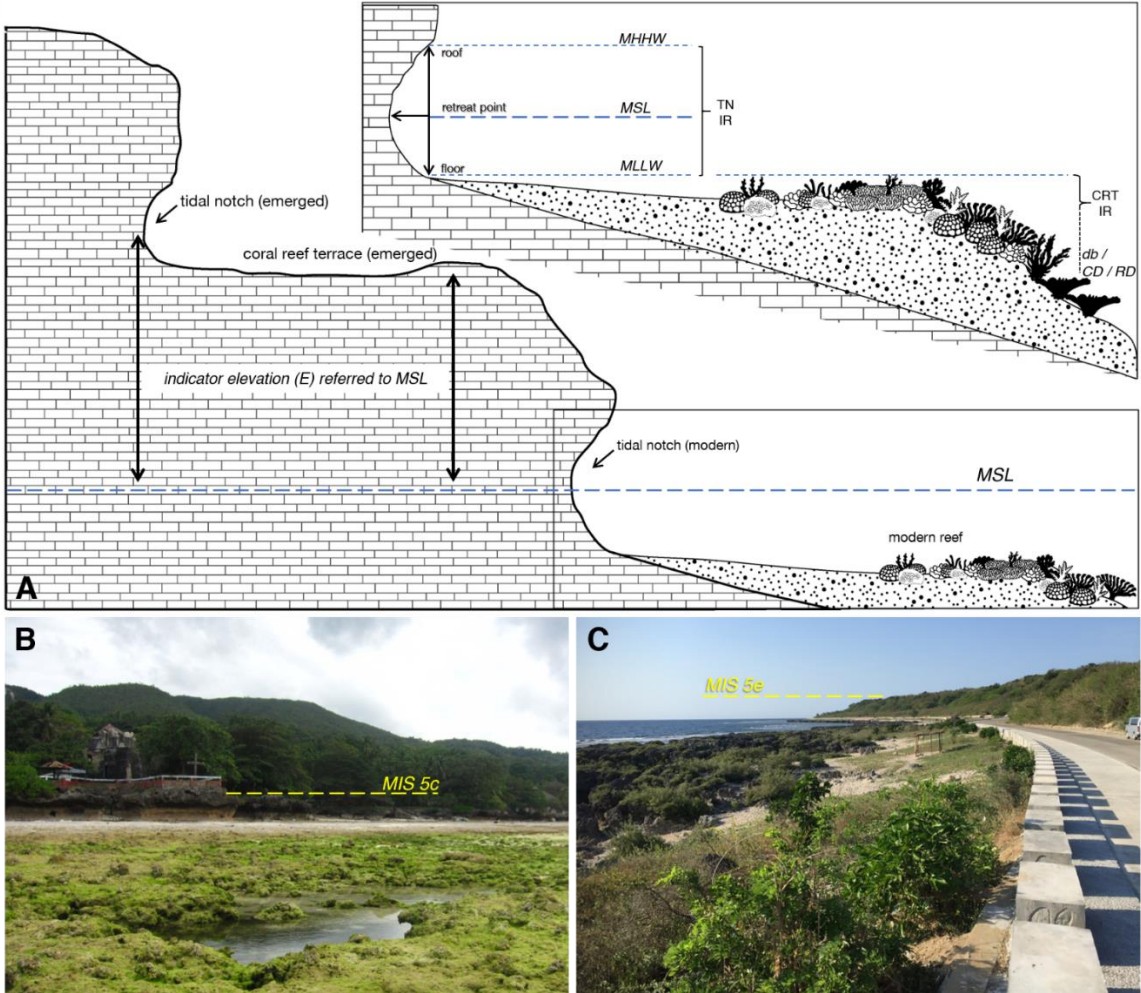

120 **Figure 2: Schematic diagram of the RSL indicators (coral reef terrace and tidal notch) and their relationship with respect to mean sea level (MSL) and representative photos of coral reef terraces in the Philippines. (A) This diagram shows the emerged RSL indicators and how their elevation (E) is measured. Shown as inset is how the indicative range (IR) was derived from the modern tidal notch (TN IR) and modern coral reef terrace (CRT IR). The upper and lower limits of the TN IR are at mean higher high water (MHHW) and mean lower low water (MLLW), respectively. The upper limit of the CRT IR is positioned at MLLW while its lower 125 limit is either at breaking depth (db) (Rovere et al., 2016; Lorscheid and Rovere, 2019) or coral depth (CD, modern coral depth distribution based on Hibbert et al., 2016) or average modern reef depth (RD). (B) A representative photo of the coral reef terrace in Punta Cruz, Bohol (Philippines) showing the trace of the MIS 5c terrace (yellow dashed line, WALIS RSL ID 1347). (C) A representative photo of the coral reef terraces in Currimao, Ilocos Norte (Philippines) showing the elevation of the identified MIS 5e terrace (yellow dashed line, WALIS RSL ID 1339).**

## 130 3.1 Coral Reef Terraces

Coral reef terraces are geomorphological sea-level indicators created by the combination of bioconstructional and erosional processes in tropical areas (Anthony, 2009; Rovere et al., 2016). Depending on the local geological and environmental settings, different elements of the reef terrace can be used to gauge the past position of sea level: the height of the paleo reef crest, the elevation of the highest in situ corals, or the average elevation of the inner margin of the reef terrace. In our database, whenever

135 available we used the elevation of the highest fossil reef crest as RSL proxy. Alternatively, we used the elevation of the highest in situ dated coral, or the inner margin of the terrace (defined as the junction between the horizontal reef platform and the terrace slope). In some cases, when no other information was available, we used the average elevation of the terrace or the average elevation of the dated sample from that terrace to attribute an elevation to the RSL indicator.

140 An important concept in estimating RSL from coral reef terraces (and from any sea-level indicator) is that of 'indicative meaning', which describes where the sea-level indicator formed with respect to paleo sea level (Shennan, 1982, Van de Plassche, 1986; Hijma et al., 2015; Rovere et al., 2016). The indicative meaning is quantified using two parameters: the 'indicative range' (IR) and the 'reference water level' (RWL). The IR represents the vertical range over which an indicator forms, while the RWL is the mid-point of this range (Hijma et al., 2015). As the indicative range increases, the uncertainty in 145 the final paleo RSL reconstruction also increases (Rovere et al., 2016). The indicative meaning for coral reef terraces is determined by measuring the relationship of the fossil coral reef terrace relative to their modern counterparts (i.e., the modern reef) with respect to present tide level (usually the mean sea level), with an associated error range.

In most of the studies reviewed, neither the indicative meaning nor the water-depth in which the modern reef is situated was 150 provided. For this reason, we used the Indicative Meaning Calculator (IMCalc) (Lorscheid and Rovere, 2019) to quantify the indicative meaning at each site. IMCalc is a stand-alone Java-app that quantifies the indicative meaning (IR and RWL) using *ex-situ* wave and tide datasets. Using IMCalc, the upper and lower limits of the modern reef is at mean lower low water (MLLW) and breaking depth of waves ($d_b$, the lowest point of interaction between sediment and water), respectively (Lorscheid and Rovere, 2019). In cases where IMCalc did not provide any value to calculate the indicative meaning (due to 155 wave datasets not available on the site of the RSL indicator), we took the MLLW from the tide datasets of IMCalc as the upper limit of the modern reef (Table 1). Typically, living corals grow up to MLLW (e.g., Rovere et al., 2016) or mean low water springs (MLWS) (e.g., Woodroffe and Webster, 2014). For the lower limit, we used the modern coral depth distribution (median water depth) of the corals sampled from the terrace using the database of Hibbert et al. (2016) or when available, we used the average reported depth of the modern reef. This may imply a wider vertical depth range depending on the coral species 160 dated from the reef terrace leaving a large indicative range for the coral reef terrace indicator.

**Table 1: Different types of RSL indicators reviewed in this study and the descriptions of their reference water level (RWL) and indicative range (IR).**

| Name of RSL indicator | Description of RSL indicator | Description of RWL | Description of IR | Indicator reference(s) |
|---|---|---|---|---|
| Coral reef terrace (general definition) | Coral-built flat surface, corresponding to shallow-water reef terrace to reef crest. The definition of indicative meaning is derived from Rovere et al., 2016, and it represents the broadest possible indicative range, that can be refined with information on coral living ranges. | (Mean Lower Low Water + Breaking depth)/2 | Mean Lower Low Water - Breaking depth | Rovere et al., 2016; Lorscheid and Rovere, 2019 |
| | | (Mean Lower Low Water + Modern coral depth distribution (median water depth) or average modern reef depth) /2 | Mean Lower Low Water - Modern coral depth distribution (median water depth) or average modern reef depth | Hibbert et al., 2016; Rovere et al., 2016; Lorscheid and Rovere, 2019 |
| Tidal notch | Definition by Antonioli et al., 2015: "Indentations or undercuttings cut into rocky coasts by processes acting in the tidal zone (such as tidal wetting and drying cycles, bioerosion, or mechanical action)". Definition of indicative meaning from Rovere et al., 2016. | (MHHW + MLLW)/2 | MHHW - MLLW | Antonioli et al., 2015 Rovere et al., 2016 |

165  **3.2 Tidal notches**

Tidal notches are "indentations or undercuttings etched or carved into rocky cliffs through various processes acting in the tidal zone such as bioerosion, wetting and drying cycles and salt weathering, hyperkarst processes, and mechanical erosion" (Antonioli et al., 2015). Tidal notches are generally used as precise indicators of paleo RSL, as their retreat point (or notch apex), which is the deepest part of the notch, usually forms near mean sea level (Pirazzoli, 1986; Antonioli et al., 2015). The

170  morphology of a tidal notch typically indicates the tidal range in an area with the notch floor and roof representing the mean low tide and mean high tide levels, respectively (Pirazzoli, 1986; Antonioli et al., 2015). The notch depth, which is the

horizontal distance between the retreat point and the projected vertical plane from the edge of the roof, is proportional to the duration of sea-level stillstands (Pirazzoli, 1986; Antonioli et al., 2015). Preservation of tidal notches above modern sea level coupled with dateable material (corals, mollusks) around the notch allows to constrain paleo sea level. To determine the indicative meaning of tidal notches, we used IMCalc to generate values for the upper limit (i.e., mean higher high water, MHHW) and lower limit (MLLW) of the tidal notch (Lorscheid and Rovere, 2019) (Table 1).

## 4. Elevation measurements and sea level datums

In the studies reviewed, different techniques were used to measure the elevation of RSL index points. These include barometric altimeter, differential GPS, theodolite and rod, total station, and auto/hand level. Some studies reported uncertainty values for the measured elevation of the sea-level indicators and these are values inserted in the database. However, when a study did not report an elevation error, we assigned a value for the vertical accuracy based on its elevation measurement technique following the typical accuracy suggested by Rovere et al. (2016) (Table 2). If the elevation measurement method was not reported by the original study, the elevation error was calculated as 20% of the reported elevation. In general, most of the studies reported elevations with respect to mean sea level (MSL). In case elevations were reported with respect to mean low water springs (MLWS), high tide level, or low tide level, we corrected the elevation to mean sea level by using the tidal datum from nearby tide stations using data from the University of Hawaii Sea Level Center (https://uhslc.soest.hawaii.edu/).

**Table 2: Measurement techniques reported by the published studies and associated measurement uncertainties.**

| Measurement technique | Description | Typical accuracy |
|---|---|---|
| Barometric altimeter | Difference in barometric pressure between a point of known elevation (often sea level) and a point of unknown elevation. Not accurate and used only rarely in sea-level studies | Up to ±20% of elevation measurement (Rovere et al., 2016); ±1 m (Major et al., 2013); ±10 m (Hantoro et al., 1994) |
| Differential GPS | GPS positions acquired in the field and corrected either in real time or during post-processing with respect to the known position of a base station or a geostationary satellite system (e.g. Omnistar). Accuracy depends on satellite signal strength, distance from base station, and number of static positions acquired at the same location. | ±0.02/±0.08 m, depending on survey conditions and instruments used (e.g., single-band vs dual-band receivers) (Rovere et al., 2016); ±0.5 m (Cox, 2009); ±2 m (Pedoja et al., 2018) |
| Not reported | The elevation measurement technique was not reported, most probably hand level or metered tape. | 20% of the original elevation reported added in root mean square to the sea level datum error |

| | | |
|---|---|---|
| | | (Rovere et al., 2016); ±5 m (Bard et al., 1996) |
| Theodolite and rod | Elevation derived from triangulation with a theodolite. | Usually very precise, centimetric accuracy, depending on distance (Rovere et al., 2016) |
| Total station or Auto/hand level | Total stations or levels measure slope distances from the instrument to a particular point and triangulate relative to the XYZ coordinates of the base station. The accuracy of this process depends on how well defined the reference point and on the distance of the surveyed point from the base station. Thus, it is necessary to benchmark the reference station with a nearby tidal datum, or use a precisely (DGPS) known geodetic point. The accuracy of the elevation measurement is also inversely proportional to the distance between the instrument and the point being measured. | ±0.1/±0.2m for total station ±0.2/±0.4 m for hand level (Rovere et al., 2016); ±1 m (Merritts et al., 1998); ±0.15 m (Ringor et al., 2004, Omura et al., 2004); ±0.10 m (Maxwell et al., 2018) |

190

## 5 Dating techniques

Carbonate skeletons such as those from corals found on top of reef terraces, if diagenetically unaltered, can be assigned a radiometric age using geochronological techniques. The ages of the LIG RSL indicators in Southeast Asia were generally determined using U-series dating of fossil corals collected from the surface or beneath reef terraces. In older studies, the U-series dating is coupled with Electron Spin Resonance (ESR) dating of corals. For this database, we inserted a total of 105 U-series ages (one of which on a mollusk), and eight ESR coral ages. Further 21 U-series coral ages were retrieved from the compilation of Chutcharavan and Dutton (2020), and assigned to RSL index points in our database. For each dated sample, we included information about whether or not the ages were originally accepted in the published studies and whether these are also cited/referred to in other studies or not. Sample details such as geographic location, taxa description, and laboratory analysis data were also inserted in the database. For the analytical details and the ages, we reported uncertainties as ±2σ. For studies reporting uncertainties as ±1σ, we standardized to ±2σ. For studies not reporting whether analytical uncertainties are given as ±1σ or ±2σ, we assumed that the given uncertainties were in ±1σ and converted them to ±2σ.

## 6 Regional overview of relative sea-level indicators

Most of the studies on LIG sea-level indicators in Southeast Asia aim to estimate long-term uplift rates and constrain vertical land movements using uplifted coral reef terraces (e.g., Chappell and Veeh, 1978, Merritts et al., 1998; Major et al., 2013; Pedoja et al., 2018). Some studies discuss LIG sea levels alongside tectonic uplift derived from these raised coral terraces (e.g., Omura et al., 2004; Ringor et al., 2004; Bard et al., 1996). In general, this database builds upon the previous compilations of Pedoja et al. (2014) and Hibbert et al. (2016). We retrieved and reviewed the original research papers mentioned in these

studies, and found additional studies that were not reported therein. We screened a total of 14 studies on LIG sea-level

210 indicators in Southeast Asia, and standardized 43 sea-level index points. A plot of these RSL proxies is presented in Fig. 3. For consistency, all site names in this database are the same as those reported in the original studies. Paleo sea level elevations cited in the following sections are in meters above mean sea level (m amsl).

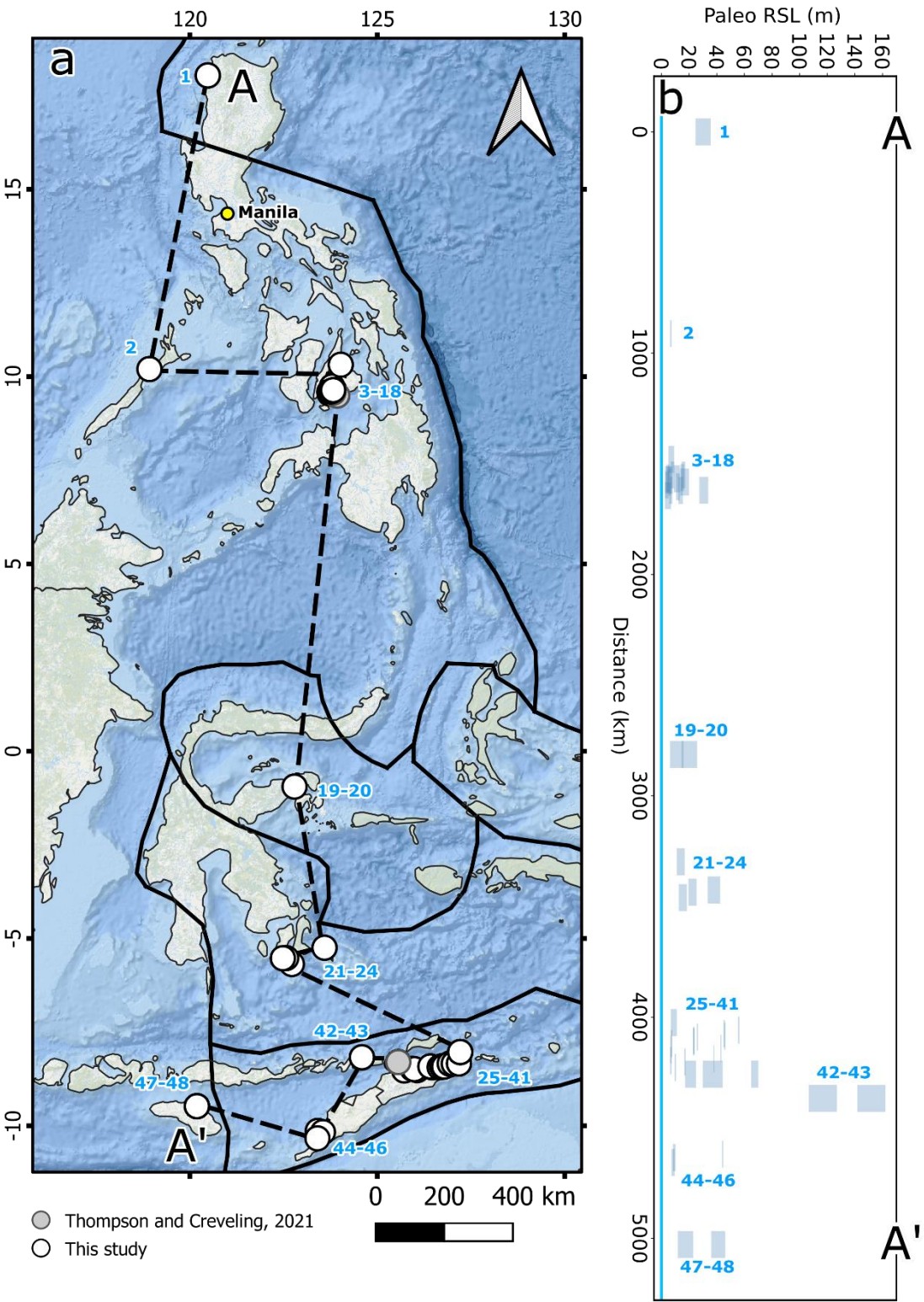

**Figure 3. MIS 5 RSL index points described in this study, plotted on a North-to-South transect (A-A'). a) Map of sites, including those reviewed by Thompson and Creveling, 2021. Solid black lines indicate tectonic plate boundaries. b) Paleo RSL versus distance along the A-A' transect shown in a). Note that this plot contains MIS 5a, 5c, and 5e datapoints. Sites list: 1: Currimao Late Pleistocene terrace (MIS 5e) (RSL ID 1339). 2: Tuturinguen Point Upper notch (MIS 5e) (RSL ID 1349). 3: Mactan Island MIS 5e terrace (RSL ID 1350). 4: Punta Cruz Upper terrace (MIS 5e) (RSL ID 1348). 5: Punta Cruz Lower terrace (MIS 5c) (RSL ID 1347). 6: Tutolan, Panglao Island MIS 5e terrace (RSL ID 1354). 7: N. of Tabalong, Panglao Is. MIS 5e terrace (RSL ID 3644). 8: N. of Bingag, Panglao Is. MIS 5e terrace (RSL ID 3643). 9: N. of Tangnan, Panglao Island MIS 5e terrace (RSL ID 3642). 10: N. of Bil-isan, Panglao Island MIS 5e terrace (RSL ID 3641). 11: San Isidro, Panglao Island Middle terrace (MIS 5e) (RSL ID 1346). 12: Panglao Island (RSL ID 3560). 13: Doljo Point, Panglao Island well-defined terrace (MIS 5e) (RSL ID 1357). 14: Taiwala, Panglao Island MIS 5e coral reef terrace (RSL ID 1355). 15: Pamilacan Island Highest terrace (MIS 5e) (RSL ID 1344). 16: Pamilacan Island MIS 5a Terrace (RSL ID 3524). 17: Pamilacan Island MIS 5c Terrace (RSL ID 3525). 18: Gak-ang, Panglao Island well-defined terrace (MIS 5e) (RSL ID 1358). 19: Luwuk, Sulawesi Lower coral reef terrace (MIS 5c) (RSL ID 3607). 20: Luwuk, Sulawesi Lower coral reef terrace (MIS 5a) (RSL ID 3633). 21: Wangi Wangi, SE Sulawesi T1 (MIS 5e) (RSL ID 3600). 22: Kadatua West, SE Sulawesi T3 (MIS 5e) (RSL ID 3605). 23: SW Buton Nirwana, SE Sulawesi T2 (MIS 5e) (RSL ID 3604). 24: SE Buton Bahari, SE Sulawesi T1 (MIS 5e) (RSL ID 3602). 25: Kisar Island Terrace I (MIS 5e) (RSL ID 3609). 26: North coast of East Timor Poros – Tutuala (Ponte Tei TIII) (RSL ID 3656). 27: North coast of East Timor Lautem-Com (Com 2 terrace) (RSL ID 3655). 28: North coast of East Timor Lautem-Com (Lautem TIV) - MIS5e (RSL ID 3654). 29: North coast of East Timor Terraces 50 km East of Baucau (Lautem area) (MIS 5a) (RSL ID 3621). 30: North coast of East Timor Laga-Buiomau (Buiomau2 TI / TII) - MIS 5a? (RSL ID 3653). 31: North coast of East Timor Laga-Buiomau (Ililai TI) - MIS 5a? (RSL ID 3652). 32: North coast of East Timor Laga-Buiomau (Laga TII) - MIS 5c? (RSL ID 3650). 33: North coast of East Timor Timor-Baucau section (MIS 5c) (RSL ID 3620). 34: North coast of East Timor Manatuto-Baucau (Liarua TI) - MIS 5e? (RSL ID 3649). 35: North coast of East Timor Timor-Manatuto section (MIS 5c) (RSL ID 3619). 36: North coast of East Timor Subau - Manatuto (Manatuto TI) - MIS 5e? (RSL ID 3647). 37: North coast of East Timor Subau - Manatuto (Subao TII) - MIS 5e (RSL ID 3645). 38: North coast of East Timor Timor-Hau section (MIS 5e) (RSL ID 3618). 39: Atauro Island Atauro Terrace 2 (MIS 5e) (RSL ID 3617). 40: Atauro Island Terrace 1a (MIS 5e) (RSL ID 3561). 41: Atauro Island Terrace 1b (RSL ID 3562). 42: Alor Island Terrace II 4 (MIS 5c) (RSL ID 3612). 43: Alor Island Terrace II6 (MIS 5e) (RSL ID 3611). 44: Cape Namosain, Kupang, West Timor Terrace 5 (MIS 5e) (RSL ID 3614). 45: Aikalui Point, Semau Island Low emergent reef (MIS 5a) (RSL ID 3613). 46: Cape Oeloimi, SE Semau Island, Kupang, W Lowest terrace (MIS 5e) (RSL ID 3615). 47: Cape Laundi, Sumba Island Reef Complex II (MIS 5e & 9) (RSL ID 3593). 48: Cape Laundi, Sumba Island Reef Complex I (MIS 5) (RSL ID 3592). Background map sources: Esri, Garmin, GEBCO, NOAA NGDC, and other contributors. Created using ArcGIS® software by Esri. ArcGIS® and ArcMap™ are the intellectual property of Esri and are used herein under license. Copyright© Esri. All rights reserved. For more information about Esri® software, please visit www.esri.com. Plate boundaries from Bird, 2003, updated by Hugo Ahlenius and Nordpil (https://github.com/fraxen/tectonicplates).**

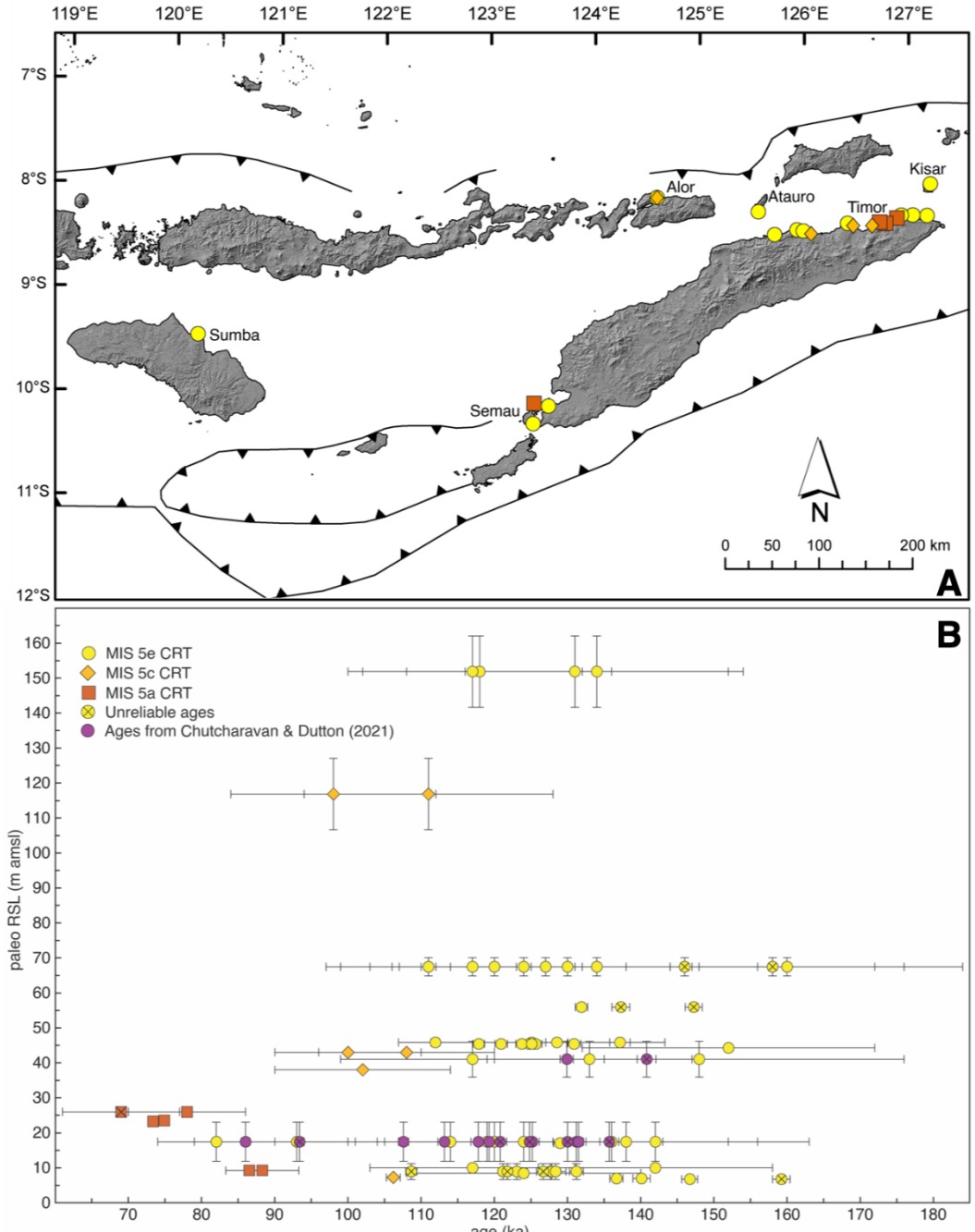

**Figure 4. RSL proxies and age-elevation plot of dated corals in Sumba-Timor-Alor region. (A)** Location map of the 22 RSL proxies screened in this study (14 of which are MIS 5e, four are MIS 5c, and four are MIS 5a). Also shown are the names of the study sites mentioned in the original publications. **(B)** Age-elevation plot of all the 87 coral samples (including unreliable ones) dated using U-series and ESR. Shown for reference are ages inserted by Chutcharavan and Dutton (2021) which are all from the coral reef terraces in Sumba Island. Digital elevation model sourced from ASTER Global Digital Elevation Model (GDEM) V003 (NASA/METI/AIST/Japan Spacesystems and U.S./Japan ASTER Science Team, 2019).

## 6.1 Sumba-Timor-Alor

The Sumba-Timor-Alor region encompasses the islands of Sumba, Timor, and Alor as well as neighboring Semau, Atauro, and Kisar Islands (Fig. 4a). This region has the highest number of RSL data points within the database, most of them reported from Timor Island. A plot of the ages of dated samples and elevation of RSL indicators is shown in Fig. 4b.

### 6.1.1 Sumba Island

A well-studied sequence of coral reef terraces was reported in Sumba Island, where fossil corals of MIS 5a, 5c, and 5e and older ages were dated (Pirazzoli et al., 1991; Pirazzoli et al., 1993; Bard et al., 1996). In Sumba Island, we compiled two RSL index points (WALIS RSL ID 3592 and 3593) from two reef complexes at Cape Laundi. For WALIS RSL ID 3592, we calculated RSL at 17.4±5.6 m from the "lower reef complex I", that was assigned by Bard et al. (1996) to a general MIS 5 age since corals belonging to MIS 5a, 5c, and 5e were dated from this reef complex. The presence of multiple sets of ages within one reef complex supports its polycyclic nature (Bard et al., 1996), at odds with earlier works (i.e., Pirazzoli et al., 1991; 1993) that designated this reef as MIS 5e based on nine coral ages (WALIS U-Series ID 2900, 2901, 2902, 2903, 2904; WALIS ESR ID 151, 152, 153, 154). For this database, we used the recent age assignment from Bard et al. (1996). WALIS RSL ID 3593 defines the "upper reef complex II", with ages corresponding to MIS 5e and MIS 9 (Bard et al., 1996). For this datapoint, we adopt the designation of Bard et al. (1996) and assigned this terrace an MIS 5e age, with RSL at 41.1±5.1 m (WALIS RSL ID 3593). For these two points, the quality of RSL data is rated as 'Poor' (2/5) due to the large vertical uncertainties, while the quality of age information is rated as 'Average' (3/5) since varying ages, from MIS 5a, 5c, 5e, and MIS 9, are reported for the same site.

### 6.1.2 Timor Island

One of the earliest works reporting LIG sea-level indicators in Southeast Asia was that of Chappell and Veeh (1978), examining coral reef terraces in East Timor (Atauro Island and north Timor Island) to assess RSL changes within the active tectonic context of the study area. A more recent work on the coral reef terraces along the northern coast of Timor Island was done by Cox (2009), to constrain the neotectonic evolution and arc-continent collision processes in the region. In their study, Cox (2009) mapped ten additional sites in Timor Island using differential GPS and provided new U-series coral ages using mass spectrometry. In Cape Namosain, situated along southwest Timor Island, Jouannic et al. (1988) reported seven steps of raised reef terraces and in the fifth step (Terrace 5, with reef crest at 44 m above low tide), a coral head in growth position was dated MIS 5e. For this study, elevations were measured using hand leveling technique and the coral age was determined using U-series dating (Jouannic et al., 1988).

In Timor Island, we standardized a total of 14 RSL points from different sites (13 of which from north coast and one from southwest coast). In total, we inserted in the database 13 RSL points from 13 reef terrace sites reported by Chappell and Veeh

(1978) and Cox (2009). Out of these 13 RSL points, seven data points are dated to MIS 5e (WALIS RSL ID 3618, 3645, 3647, 3649, 3654, 3655, 3656). The highest MIS 5e paleo RSL is at an elevation of 55.9±0.5 m (WALIS RSL ID 3656), and is derived from the highest dated coral sampled from a terrace at Poros – Tutuala site. While the quality of RSL information for this point is rated as 'Excellent' (5/5), its age quality is rated 'Average' (3/5) since the three ages from this site are just above

the upper limit of MIS 5e (132-147 ka, WALIS U-Series IDs 2976, 2978, 2979). The lowest MIS 5e RSL datapoint is at 6.7±0.5 m (WALIS RSL ID 3647) at the Subau - Manatuto section, with the same quality scores as the previous site, since the two ages from this site are above the upper limit of MIS 5e (147-159 ka, WALIS U-Series ID 2966 and 2977).

Three MIS 5c RSL points (WALIS RSL ID 3619, 3620, 3650) were extracted from Timor-Manatuto, Timor-Baucau, and

Laga-Buiomau sites. The highest MIS 5c RSL is found at 43±0.4 m, in correspondence of a reef terrace at Baucau area (WALIS RSL ID 3620). For this point, the quality of RSL data is Excellent (5/5) and the quality of age information is Good (4/5). Three MIS 5a RSL index points (WALIS RSL ID 3621, 3652, 3653) are identified from Lautem, Laga-Buiomau (Ililai), and Laga-Buiomau (Buiomau) sites. The highest MIS 5a RSL is located at 26±0.4 m (WALIS RSL ID 3621) in the Lautem area, east of Baucau.


In the southwestern coast of Timor Island, in Cape Namosain, we compiled one RSL data point (WALIS RSL ID 3614) derived from a reef terrace reported by Jouannic et al. (1988). The MIS 5e paleo sea level at this point is 44.3±0.4 m (WALIS RSL ID 3614) with quality of RSL data rated as 'Excellent' (5/5) and quality of age information as 'Good (4/5).

### 6.1.3 Alor Island

A flight of six coral reef terraces in Alor Island, located offshore north of Timor Island, were studied by Hantoro et al. (1994) and dated with U-series and ESR techniques. In this study, terrace elevations (Terrace $II_6$ at 165±10 m and Terrace $II_4$ at 115±10 m) were measured using pocket altimeters, with an accuracy of ±10 m (Hantoro et al., 1994). Two RSL data points were extracted from coral reef terraces in Alor Island studied by Hantoro et al. (1994). These terraces were dated MIS 5e and 5c, with the former being represented by the well-developed complex of Terraces $II_5$ and/or $II_6$, and the latter being represented

by Terrace $II_4$ (Hantoro et al., 1994). The first datapoint was formed with a paleo RSL at 151.9±10.2 m (WALIS RSL ID 3611, MIS 5e). The second RSL point is derived from the MIS 5c reef terrace and marks paleo RSL at 116.9±10.2 m (WALIS RSL ID 3612). For both points, the quality of RSL data is rated as 'Poor' (2/5) due to the high vertical uncertainty and the quality of age information is rated as 'Good' (4/5). The calculated MIS 5e and MIS 5c paleo sea levels in this island are the highest in the Southeast Asian region.

### 6.1.4 Semau Island

In Semau Island, we extracted two RSL data points from the coral reef terraces studied by Merritts et al. (1998) and Jouannic et al. (1988). In Cape Oeloimi, situated at the southeast coast of the island, a reef terrace dated MIS 5e is reported by Jouannic

et al. (1988) while in Aikalui Point, located in the north coast of the island, a low emergent reef is dated MIS 5a by Merritts et al. (1998). The point represent a paleo RSL at 8.4±1.3 m (WALIS RSL ID 3615), and is dated to MIS 5e. The second datapoint

marks RSL at 9.2±1.1 m (WALIS RSL ID 3613) and is dated to MIS 5a. For both RSL indicators, both the quality of RSL and age information are rated as 'Good' (4/4).

### 6.1.5 Atauro Island

In Atauro Island, two RSL data points were derived from the coral reef terraces studied by Chappell and Veeh (1978). In this site, seven steps of coral reef terraces have been identified, but only the two lowest ones have been dated. The lowest terrace,

Terrace 1b, is dated to MIS 5c and Terrace 2 is dated to MIS 5e (Chappell and Veeh, 1978). Two index points at Atauro Island were already standardized into the WALIS template by Thompson and Creveling (2021). These correspond to Terrace 1a (WALIS RSL ID 3561) and Terrace 1b (WALIS RSL ID 3562, dated to MIS 5c with U-Series). To these datapoints, we added the terrace dated to MIS 5e at 67.49± 2.6 m (WALIS RSL ID 3617), that was also dated with U-Series. For this point, the quality of RSL data is 'Average' (3/5) while the quality of age information is 'Excellent' (5/5) since several dated samples (12

coral ages) clearly point to a MIS 5e.

### 6.1.6 Kisar Island

In Kisar Island, multiple uplifted terraces of coralline limestone (Terrace I to V) were studied by Major et al. (2013). The terrace elevations were measured using barometric altimeters and samples (corals and mollusk) were dated using U-series dating (Major et al., 2013). The lowest terrace, Terrace I, was dated to MIS 5e, and indicates RSL at 8.9±2.3 m (WALIS RSL

ID 3609). For this data point, we rated the quality of RSL information as 'Average' (3/5) and that of age information as 'Good' (4/5) since one reliable age and several referable (when the initial $\delta^{234}U$ is slightly out of expected range and showing evidence for some alteration but not enough to affect the age significantly; see Major et al. (2013)), samples point to an MIS 5e age.

### 6.2 Sulawesi

The Sulawesi region encompasses sites from Luwuk and southeast Sulawesi (Fig. 5a). In this region, we standardized six RSL

index points. A plot of the ages of dated samples and elevation of RSL indicators is shown in Fig. 5b.

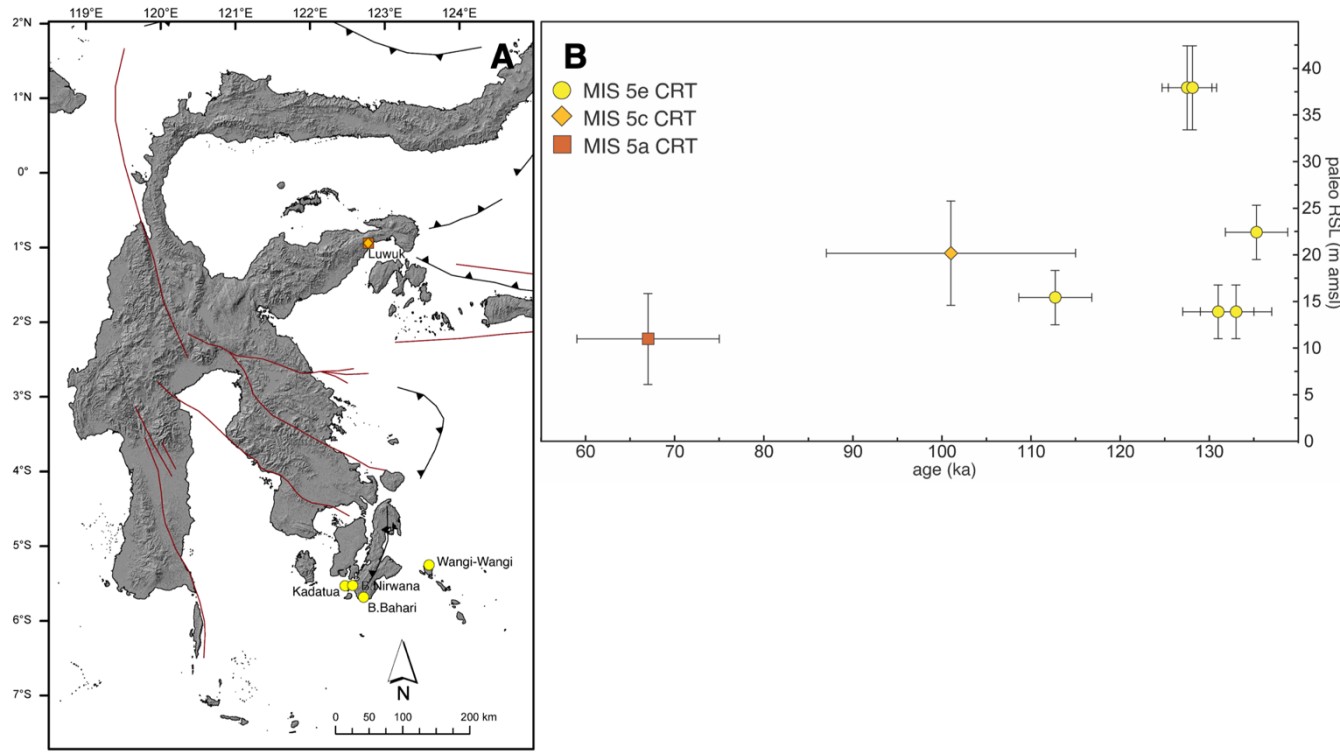

**Figure 5. RSL proxies and age-elevation plot of dated corals in Sulawesi. (A) Location map of the six RSL proxies screened in this study (four of which are MIS 5e from southeast Sulawesi, the remaining two points are of MIS 5a and MIS 5c and are from Luwuk). Also shown are the names of the study sites mentioned in the original publications. (B) Age-elevation plot of all the eight coral samples dated using U-series. Digital elevation model sourced from ASTER GDEM V003 (NASA/METI/AIST/Japan Spacesystems and U.S./Japan ASTER Science Team, 2019).**

### 6.2.1 Luwuk

In central Sulawesi, in the area of Luwuk, coral reef terraces were studied by Sumosusastro et al. (1989), who, however, did not describe the measurement technique they employed to measure the elevations of the reef terrace. In Luwuk, the reef terraces were divided into a lower, a middle, and an upper group and only the lower terraces are dated (Sumosusastro et al., 1989). Two coral samples yielded ages of MIS 5c (WALIS U-Series ID 2911) and MIS 5a (WALIS U-Series ID 2913) allowing us to extract two RSL index points for this site. The first RSL data point represents MIS 5c RSL at 20.2±5.6 m (WALIS RSL ID 3607), while the second is dated to MIS 5a and represents RSL at 11.0±4.9 m (WALIS RSL ID 3633). For both points, the quality of RSL data is rated as 'Poor' (2/5) due to the large vertical uncertainties and the quality of age information is rated as 'Good' (4/5). No MIS 5e paleo sea level was retrieved in the Luwuk area.

### 6.2.2 Southeast Sulawesi

In southeast Sulawesi, coral reef terraces at four sites (Wangi-Wangi, Buton Bahari, Buton Nirwana, Kadatua) were dated to MIS 5e (Pedoja et al., 2018). From these sites, we extracted four RSL data points (WALIS RSL ID 3600, 3602, 3604, 3605).

The highest MIS 5e RSL is located at 38.0±4.5 m (WALIS RSL ID 3605), in correspondence of Terrace 3, at Kadatua site. The quality of RSL data for this site is rated as 'Poor' (2/5) due to the large vertical uncertainty on the final RSL calculation, while the quality of age information is rated as 'Excellent' (5/5) since two U-series ages (WALIS U-Series ID 1264 and 1265) for this data point clearly point to MIS 5e. For Wangi-Wangi site, the MIS 5e RSL is located at an elevation of at 13.9±2.9 m

(WALIS RSL ID 3600). At Buton Bahari, the MIS 5e RSL is situated at 15.4±2.9 m (WALIS RSL ID 3602). At Buton Nirwana site, the MIS 5e RSL is situated at 22.4±2.9 m (WALIS RSL ID 3604). For these three sites, the quality of RSL data is rated as 'Average' (3/5) while the quality of age information is rated as 'Good (4/5).

## 6.3 The Philippines

The rocky coastlines of the Philippines are mostly fringed by coral reef terraces. The sea-level indicators in the Philippines

(coral reef terraces and tidal notches) were generally examined to constrain Holocene paleo sea level (e.g., Maeda et al., 2004; Maeda and Siringan, 2004; Shen et al., 2010) and derive tectonic uplift rates (e.g., Ramos and Tsutsumi, 2010; Maxwell et al., 2018). Three studies report on the elevations of LIG sea-level indicators and provide coral ages (Omura et al., 2004; Ringor et al., 2004; Maxwell et al., 2018). Geomorphic mapping of the raised coral reef terraces in west Luzon by Maemoku and Paladio (1992) lacks radiometric age constraints and was not included in the WALIS database. The elevations of the LIG sea-level

proxies in the Philippines were measured using various techniques (electronic distance measuring devices, hand levelling, and laser rangefinder and rod) with generally high accuracies (±0.3–10 cm). Elevation data were usually corrected and normalized to the present mean sea level. Ages were determined using U-series dating of fossil corals collected from the surface or beneath LIG reef terraces. Sites studied in the Philippines are: the islands of Bohol, Cebu, and Palawan and the area of Ilocos Norte (Fig. 6a). In this region, we standardized 15 RSL points, most of them located in Bohol Island. Out of the 15 RSL points, 14

data points are derived from coral reef terraces while one point is from tidal notch data. In addition, three RSL points were already inserted by Thompson and Creveling (2021) and we refer to their points in the discussion below. A plot of the ages of dated samples and elevation of RSL indicators is shown in Fig. 6b.

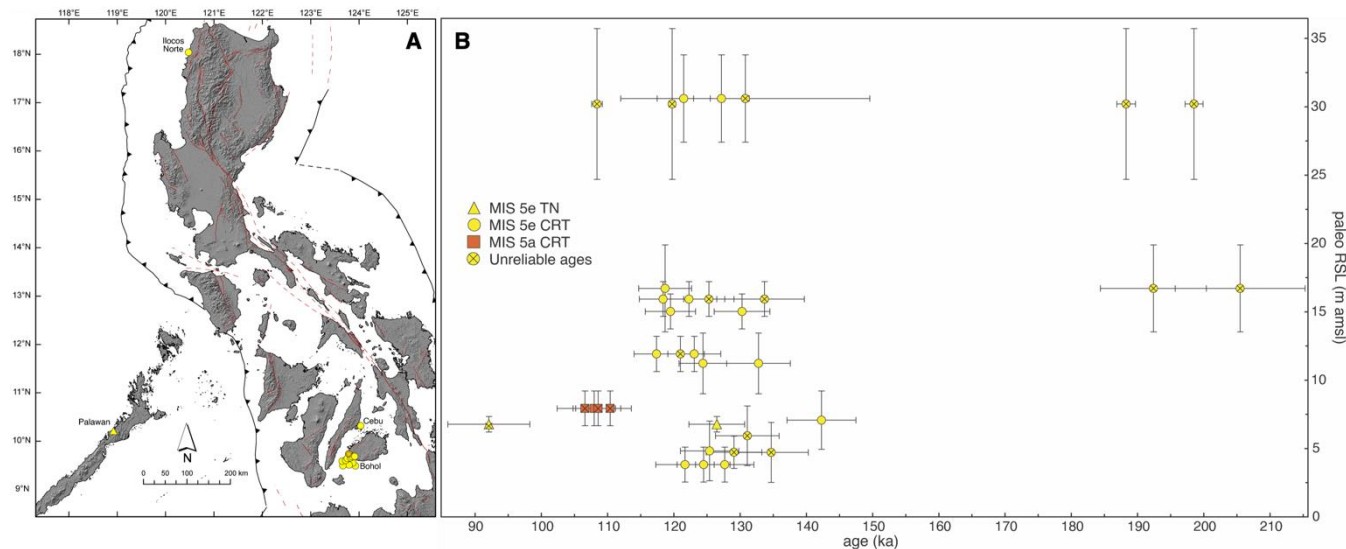

**Figure 6. RSL proxies and age-elevation plot of dated corals in the Philippines. (A) Location map of the 15 RSL proxies screened in this study (13 of which from MIS 5e coral reef terraces (CRT), one from MIS 5e tidal notch (TN), and one from MIS 5c CRT). Also shown are the names of the study sites mentioned in the original publications. (B) Age-elevation plot of all the 35 coral samples (including unreliable ones) dated using U-series. Digital elevation model sourced from ASTER GDEM V003 (NASA/METI/AIST/Japan Spacesystems and U.S./Japan ASTER Science Team, 2019).**

### 6.3.1 Bohol

In Bohol province, most notably on the Panglao and Pamilacan islands, a series of coral reef terraces were dated to MIS 5a, 5c, and 5e (Omura et al., 2004; Ringor et al., 2004). In these two islands, three terraces were already inserted in WALIS by Thompson and Creveling (2021). Two are located at Pamilacan Island, and are dated to MIS 5a and MIS 5c (WALIS RSL IDs respectively 3524 and 3525), and one is located at San Isidro (Panglao Island), with MIS 5c age (WALIS RSL ID 3560).

The most notable site in Bohol province was reported at Pamilacan Island, a small island located southwest of Bohol Island (Ringor et al., 2004). Here, there is a complete sequence of MIS 5e, 5c, and 5a coral reef terraces. The MIS 5e RSL is at 30.6±3.2 m (WALIS RSL ID 1344), while MIS 5c and MIS 5a paleo RSL reported by Thompson and Creveling (2021) are, respectively, at 13.9±1.6 (WALIS RSL ID 3525) and 6.9±1.0 (WALIS RSL ID 3524).

Two other sites in the Bohol province where MIS 5e and MIS 5c terraces were reported (Omura et al., 2004; Ringor et al., 2004) are Punta Cruz (Bohol Island) and San Isidro (Panglao Island). In Punta Cruz, two steps of coral reef terraces were recognized, the lower at 6 m and the upper at 14 m above present sea level (Omura et al., 2004; Ringor et al., 2004). From the lower terrace, fossil corals yielded MIS 5e and 5c ages, although the MIS 5c ages are deemed less reliable (Ringor et al., 2004). Due to the difficulty of finding suitable coral samples for dating, Ringor et al. (2004) assigned an MIS 5e age to the upper terrace based on the occurrence of MIS 5c corals beneath it and a *Porites* sampled from a 25-m high terrace located 1 km north of this site which dated 246±7.5 ka. For this database, we used the age designation of Ringor et al. (2004) and assigned an MIS

5e age to the upper terrace and an MIS 5c age to the lower terrace. In Punta Cruz, the MIS 5e paleo sea level is at 15.9±1.3 m (WALIS RSL ID 1348) while the MIS 5c paleo sea level is at 7.9±1.3 m (WALIS RSL ID 1347). The quality of RSL data for both RSL data points is rated as 'Good' (4/5). For which concerns the quality of age information, the MIS 5c index point is rated as 'Good' (4/5), while the MIS 5e one is rated as 'Poor' (2/5) since the age determination of the MIS 5e terrace was complicated by the difficulty of finding suitable coral samples for dating.

In San Isidro, three steps of coral reef terraces were recognized (Omura et al., 2004; Ringor et al., 2004). The middle terrace at 12.9 m above sea level was assigned to MIS 5e, while the lowest terrace was inserted in WALIS by Thompson and Creveling (2021) at 5 m above present sea level, and was assigned to MIS 5c (Omura et al., 2004; Ringor et al., 2004, WALIS RSL ID 3560). The MIS 5e RSL, derived from the middle terrace, is located at 16.7±3.2 m (WALIS RSL ID 1346) with quality of RSL data rating of 'Poor' (2/5) due to large vertical uncertainties, and quality of age rating of 'Average' (3/5). The remaining eight RSL data points around Panglao Island are obtained from MIS 5e coral reef terraces (WALIS RSL ID 1354, 1355, 1357, 1358, 3641, 3642, 3643, 3644) and the highest MIS 5e RSL from these points is situated at 15.0±1.3 m and is derived from a reef terrace in Tutolan (WALIS RSL ID 1354). The quality of RSL information and age for this datapoint is rated as 'Good' (4/5).

### 6.3.2 Cebu

In the Cebu province, in Mactan Island, one RSL data point is extracted from a coral reef terrace studied by Omura et al. (2004). For this site, one RSL data point is derived from a fossil coral collected atop a coral reef terrace which yielded an age of 142.3±5.2 ka (WALIS U-Series ID 2018) suggesting that an MIS 5e terrace is present in the island. For this site, the MIS 5e RSL is situated at 7.1±2.1 m (WALIS RSL ID 1350) with quality of RSL information rating as 'Average' (3/5) and that of age information as 'Good' (4/5).

### 6.3.3 Palawan

In Palawan, at Tuturinguen Point, well-developed tidal notches (with elevations at 1.5 m and 6.8 m above present sea level) are etched into pre-Quaternary limestones (Omura et al., 2004). The upper notch is dated to MIS 5e based on U-series dating of a fossil *Goniastrea* coral sample taken from a calcareous crust that drapes its floor (Omura et al., 2004). For this site, the MIS5e RSL is situated at 6.8±0.6 m (WALIS RSL ID 1349). We assigned a quality rating of 'Excellent' (5/5) to this site since the retreat point of the notch represents a precise indicator of sea level and the U-series age of a coral sample clearly connected to it points to MIS 5e (126.5±4.2 ka, WALIS U-Series ID 2020).

### 6.3.4 Ilocos Norte

In Ilocos Norte, northwest Luzon, Maxwell et al. (2018) recognized Late Pleistocene coral reef terraces. At one particular site, in Currimao, four coral samples yielded less reliable ages (108-198 ka, WALIS U-Series ID 2049, 2050, 2051, 2052). Despite

not being precise, these ages provide evidence for an MIS5e terrace in Currimao, Ilocos Norte (Maxwell et al., 2018). For this site, we inserted one RSL data point, with RSL calculated at 30.2±5.5 m (WALIS RSL ID 1339). For this RSL point, the

quality of RSL data is rated as 'Poor' (2/5) due to the large paleo sea level uncertainty, while the quality of age information is rated as 'Average' (3/5) since there is less confidence on the accuracy of the ages at this site.

## 7 Further details on paleo sea level in Southeast Asia

### 7.1 LIG sea-level fluctuations

Our Southeast Asia database provides a standardized picture of the distribution of previously published LIG sea-level proxies

in the region. In general, the examination of LIG RSL indicators in Southeast Asia revolves around understanding paleo sea level within the context of a tectonically active region (Fig. 3). Reliable ages are generally provided based on the U-series dating of corals collected on top or beneath the terrace tracts. Constraining the precise timing of sea-level changes is complicated by the difficulty of finding pristine corals for dating, and the presence of varying coral ages within a reef terrace. Bard et al. (1996) highlighted the polycyclic nature of coral reef terraces in which MIS 5a, MIS 5c, and MIS 5c corals can be

documented within one reef complex. This adds complications to the use of coral reef terraces in constraining the precise timing of LIG RSL paleo sea level within a region. Aside from that, the effects of tectonics on the LIG sea level should be taken into consideration in the examination of LIG sea-level fluctuations since the region is marked by various collision and subduction processes occurring since the Cenozoic (Hall, 2012). This region, however complicated, is still an ideal place to examine the stratigraphic and geologic record of coral reef terraces especially now that more reliable dating and accurate

elevation measurement techniques are becoming more available.

### 7.2 Sea-level indicators from other interglacials

Some of the studies that investigated and dated the LIG coral reef terraces also reported and dated older terraces at higher elevations (Sumosusastro et al., 1989; Pirazzoli et al., 1991; Pirazzoli et al., 1993; Hantoro et al., 1994; Bard et al., 1996; Omura et al., 2004; Ringor et al., 2004; Pedoja et al., 2018). Some studies infer that the higher terraces are correlated with

older interglacials although no geochronological constraint was provided (e.g., Chappell and Veeh, 1978; Hantoro et al, 1994). One of the classic examples of well-dated coral reef terraces extending up to mid-Pleistocene (ca. one million years ago) is the sequence of raised coral reef terrace in Sumba Island (Pirazzoli et al., 1991; 1993). The terraces in this site were dated using U-series and ESR dating of corals and this allowed Pirazzoli et al. (1991; 1993) to constrain the terraces from MIS 5 to MIS 29. Bard et al. (1996) dated the uplifted coral reef sequence in Sumba Island using mass spectrometry and identified terraces

corresponding to MIS 5a, MIS 5c, MIS 5e, and MIS 9. In southeast Sulawesi, coral reef and marine terraces dated MIS 7 and 9 were reported by Pedoja et al. (2018) on the basis of U/Th coral ages. In Luwuk, Sulawesi, a coral sample collected from an upper group of terraces situated at 410 to 418 m above sea level yielded a U-Th age of 229 ka (Sumosusastro et al., 1989). Hantoro et al. (1994) reported coral reef terraces at higher elevations up to 580 m in Alor Island and attempted to date an in

situ *Platygyra* coral using ESR. This sample, situated 280 m above low tide level, yielded an ESR age of 192±48 ka and terrace
elevations at 200-290 m were then correlated to MIS 7 (Hantoro et al., 1994). Higher terraces were correlated to MIS 9, 11, and 13 based on their morphology and height (Hantoro et al., 1994). In Bohol, central Philippines, terraces belonging to MIS 7 have been dated using U-series dating of corals (Omura et al., 2004; Ringor et al., 2004).

### 7.3 Holocene sea-level indicators

Aside from MIS 5e RSL indicators that are the focus of this study, Holocene RSL indicators are also documented in Southeast
Asia. In fact, studies on Holocene RSL indicators in the region are far more abundant than those on LIG RSL indicators. Previous papers that have provided an initial compilation of Holocene sea-level data for Southeast Asia include Woodroffe and Horton (2005) and Mann et al. (2019). Efforts in compiling the Holocene RSL indicators within a standardized framework are currently being done through the HOLSEA working group (Khan et al., 2019; https://www.holsea.org).

### 7.4 Uncertainties and data quality

The data reported in this compilation is subject to a variety of uncertainties that are related to the originally reported elevation and ages of the published studies. The final paleo sea level and uncertainty calculated in this database is derived from careful review of the sea-level indicators reported in the previous studies and assignment of indicative meaning values. The procedures in dealing with the elevation and age uncertainties were discussed in the previous sections on elevation measurements and dating techniques. The quality rating of RSL data points compiled in this database (quality of RSL data and quality of age
information) followed the WALIS evaluation guidelines (Rovere et al., 2020).

## 8 Future research directions

The present standardized database of LIG sea-level indicators in SE Asia compiles available studies on coral reef terraces and tidal notches in the Philippines, Sulawesi, and Sumba-Timor-Alor regions. While reliable ages were obtained by U-Th-dating of fossil corals, challenges still persist due to preservation potential of corals and the difficulty of finding pristine,
diagenetically unaltered and thus dateable samples. The calculation of paleo sea level from coral reef terraces is difficult since vertical uncertainties from measurement techniques are not mentioned in many cases and the indicative meaning of the sea-level indicators is not well documented. Precise elevation measurements (of the sea-level indicators and their modern counterparts) and finding pristine samples for dating will definitely improve the quality of these RSL proxies. One recommendation for future studies is to also measure the indicative meaning of the modern analogs and to include all sources
of vertical uncertainty.

For this compilation, most of the LIG (MIS 5e) sea-level indicators were found in the eastern portion of Southeast Asia, which is situated in the direct vicinity of major tectonic plates. We did not encounter available publications on LIG sea-level indicators in the western portion of Southeast Asia, which lies inside the subsiding Sundaland block. A tidal notch from Palawan Island in the Philippines, which lies on the eastern boundary of the Sundaland block, reveals MIS5e RSL at 6.8±0.6 m (WALIS RSL

ID 1349). The presence of an emergent LIG sea-level indicator within its eastern border as opposed to the lack of evidence of higher-than-present sea level within its western portion (e.g., mainland Southeast Asia, peninsular Malaysia, Sunda shelf) reveals that the Sundaland block might have also experienced varying sea-level histories from site to site. To understand this better, it is worthwhile to re-visit previously studied sites and explore select sites in the Sundaland block (e.g., mainland Southeast Asia) for future work. Documenting emergent LIG sea-level indicators in these locations might help unravel the

long-term tectonic behavior of this region.

  In this paper, we did not include a discussion on how the elevation of paleo RSL proxies links with uplift rates across Southeast Asia, as these are endeavors that go beyond the description of our database. However, we remark that the paleo sea level estimates we report are still uncorrected for post-depositional land movements (e.g., local or regional tectonic effects, sediment compaction, dynamic topography) or glacial isostatic adjustment (GIA) effects. Future studies might be directed towards

exploring the magnitude of GIA in this region and using our data to gather the magnitude of other post-depositional movements, comparing them with datasets from structural geology (e.g., fault slip rates and earthquake activity) or even from short-term datasets, such as land motion rates derived from GPS datasets.

## 9 Data availability

  The compiled database for Southeast Asia is available here: https://doi.org/10.5281/zenodo.4681325 (Maxwell et al., 2021).

The description of the database fields can be found here: https://doi.org/10.5281/zenodo.3961544 (Rovere et al., 2020). More information on the World Atlas of Last Interglacial Shorelines can be found here: https://warmcoasts.eu/world-atlas.html. We encourage the users of this database to cite the original sources in addition to our database and this article.

## Author contributions

  KM was the primary author of the paper and responsible for all entries into WALIS. HW and AR contributed to the structure

and writing of the manuscript and discussions on compiling the database.

## Competing interests

  The authors declare that they have no conflict of interest.

## Special issue statement

  This article is part of the special issue "WALIS – the World Atlas of Last Interglacial Shorelines". It is not associated with a

conference.

**Acknowledgements**

The authors acknowledge the inspiring environment of the doctoral program Marie Skłodowska-Curie Innovative Training Network (ITN) 4D-REEF, coordinated by Willem Renema. Discussions with members of the working group Geoecology and Carbonate Sedimentology at ZMT and U-series data entry assistance by P. Chutcharavan contributed in the refinement of this
paper. The data presented in this study were compiled in WALIS, a sea-level database interface developed by the ERC Starting Grant "WARMCOASTS" (ERC-StG-802414), in collaboration with PALSEA (PAGES / INQUA) working group. The database structure was designed by A. Rovere, D. Ryan, T. Lorscheid, A. Dutton, P. Chutcharavan, D. Brill, N. Jankowski, D. Mueller, M. Bartz, E. Gowan, and K. Cohen. The data points in this study were contributed to WALIS by K. Maxwell. This publication is part of K.M.'s doctoral thesis.

**Financial support**

This research was supported by the program 4D-REEF funded from the European Union's Horizon 2020 research and innovation programme under the Marie Sklodowska-Curie grant agreement No 813360 to H.W. A.R. received financial support from the ERC Starting Grant "WARMCOASTS" (ERC-StG-802414).

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
