# Peer review of "A standardized database of Last interglacial (MIS 5e) sea-level indicators in Southeast Asia"

_Earth System Science Data, 2021_

## Author Comment (AC1)

*Professor Matteo Vacchi*
**Editor**
WALIS – the World Atlas of Last Interglacial Shorelines
Earth System Science Data

Dear *Prof. Vacchi*,

Please find enclosed the revised manuscript entitled "**A standardized database of Last interglacial (MIS 5e) sea-level indicators in Southeast Asia**" to be considered for publication in the Special Issue "WALIS – the World Atlas of Last Interglacial Shorelines" under the Earth System Science Data journal.

This work provides a standardized database on the LIG sea-level indicators in Southeast Asia. We screened and reviewed 14 published studies on LIG RSL proxies in Sumba-Timor-Alor region, Sulawesi, and the Philippines and identified 43 unique RSL proxies (42 coral reef terraces and one tidal notch), that were correlated to 134 dated samples. Five data points date to MIS 5a (80 ka), six data points are MIS 5c (100 ka), and the rest are dated to MIS 5e.

We express our sincerest gratitude to the editor and to the two reviewers who gave their time and effort in thoroughly reviewing our manuscript and providing very useful insights and constructive comments. We accepted most of the suggestions and comments of the reviewers, which you will find addressed in the attached point-by-point response to the reviewers. The only suggestion we did not accept was to expand too much on the tectonic rates that can be derived from our data. We think that doing this would go beyond the scope of both paper and journal. We rely on your advice on this matter.

This paper presents our original unpublished work and is not under review for publication elsewhere. We also declare no conflict(s) of interests.

With our best regards,

Kathrine Maxwell*, Hildegard Westphal, Alessio Rovere
*corresponding author

**Responses to the Comments of the Reviewers**

**Manuscript No. essd-2021-126**
**A standardized database of Last interglacial (MIS 5e) sea-level indicators in Southeast Asia**

First, we would like to thank Anonymous Referee #1 and Dr. Gino de Gelder for dedicating their time in thoroughly reviewing our manuscript. We thank you for your helpful insights, constructive comments, and thorough review, which helped improve the quality of our manuscript. We accepted most of the suggestions and revised the text accordingly. Please find below our responses to the comments and corresponding pages in the manuscript where revisions appear. The original comments from the reviewers are in italics, our response is in standard font, and changes on the revised manuscript are shown in blue text.

**Responses to the comments of Reviewer #1 (RC1 comments)**
**Citation**: https://doi.org/10.5194/essd-2021-126-RC1

*This is an interesting dataset that screened and reviewed 14 studies on LIG sea-level indicators in Southeast Asia. After screening the authors report 43 unique RSL proxies that are predominantly coral reef terraces along with data point from a tidal notch. The sites are collated against 134 dated samples from U-series dating and some Electron Spin Resonance (ESR) dating from older studies.*

*The work is publishable and provides a good summary of the dataset and some implications for future work.*

*I have one primary criticism of the work and that relates to the discussion or lack thereof regarding tectonics. The authors go to great lengths to describe the tectonics of the region in the first part of the paper and this is a great start. They are indeed correct inferring that almost all data from Southeast Asia will be subject to tectonic contamination and they summarise this nicely at the start.*

*The authors then say very little about tectonics when they start presenting the datasets and plotting the data. This is shame because if one was to assume a LIG global sea level high of somewhere between 5-10 metres (there are many to choose from) and transpose that to their figures then the influence of tectonics since the LIG may be somewhat constrained in terms of average uplift rates. They could then compare that to modern uplift rates in the same systems.*

Response: We thank Anonymous Referee #1 for the insightful review. We included a discussion of the tectonic setting of Southeast Asia to give the readers an overview of general tectonics of the region. For this data paper, we did not include a discussion on how these paleo sea-level estimates can be used to derive average uplift rates as this is beyond the scope of this paper and this journal. The paleo sea level we compiled is still uncorrected for post-depositional land movements (e.g., local or regional tectonic effects, sediment compaction, dynamic topography) or glacial isostatic adjustment (GIA) effects; doing so would entail a more detailed study which is no longer within the scope of our data paper. We encourage future researchers to use the standardized data we compiled in their examination of the effects of post-depositional land movements or GIA in Southeast Asia. To address this comment, we included an explanation of the scope of the paper in Section 8: Future research directions (Line 466).

Line 488: In this paper, we did not include a discussion on how the elevation of paleo RSL proxies links with uplift rates across Southeast Asia, as these are endeavors that go beyond the description of our database. However, we remark that the paleo sea level estimates we report are still uncorrected for post-depositional land movements (e.g., local or regional tectonic effects, sediment compaction, dynamic topography) or glacial isostatic adjustment (GIA) effects. Future studies might be directed towards exploring the magnitude of GIA in this region and using our data to gather the magnitude of other post-depositional movements, comparing them with datasets from structural geology (e.g., fault slip rates and earthquake activity) or even from short-term datasets, such as land motion rates derived from GPS datasets.

*Minor comments include a few areas of awkward wording and some inclusions which seem superfluous.*

*Examples include:*

*Line 36 – Despite our .... (suggest rewording this sentence)*

R: We edited the manuscript to observe proper syntax.

Line 38: Despite our work mostly being aimed at compiling MIS 5e data, we also inserted MIS 5a and MIS 5c sites whenever they were not already in the database at the time of compilation.

*Line 80 – vague statement. Please clarify and reword*

R: We edited this statement accordingly.

Line 84: At present, the deformation in the Philippines, as revealed through the dense seismicity data since the 1900s (Philippine Institute of Volcanology and Seismology - PHIVOLCS, 2016), is predominantly controlled by movement along tectonic structures such as subduction zones and numerous onshore and offshore faults.

*Line 435 – I am not sure I see the point of mentioning the Holocene database in section 7.3? If it is as a template for further studies maybe make that explicit.*

R: We addressed this comment by revising Section 7.3 to explicitly mention that aside from MIS 5 and older RSL indicators, Holocene RSL indicators are also documented in the region.

Line 453: Aside from MIS 5e RSL indicators that are the focus of this study, Holocene RSL indicators are also documented in Southeast Asia. In fact, studies on Holocene RSL indicators in the region are far more abundant than those on LIG RSL indicators. Previous papers that have provided an initial compilation of Holocene sea-level data for Southeast Asia include Woodroffe and Horton (2005) and Mann et al. (2019). Efforts in compiling the Holocene RSL indicators within a standardized framework are currently being done through the HOLSEA working group (Khan et al., 2019; https://www.holsea.org).

**Responses to the comments of Reviewer #2 (RC2 comments)**
**Citation**: https://doi.org/10.5194/essd-2021-126-RC2

*This paper presents a compilation of SE-Asian relative sea-level indicators from Marine Isotope Stage (MIS) 5e, and some from MIS 5a and 5c. Following a brief introduction, the authors give some background on SE-Asian tectonics, sea-level markers and technical details of the compilation, before going into more details on the specific sites compiled. The paper ends with some general discussion points and future research directions.*

*I think the paper is well written, and the authors have done a thorough job in systematically and consistently compiling all the data, which was undoubtedly a big effort. This will surely serve as a useful basis for researchers working on Quaternary sea-level in the region (including myself). As with any data-focused paper, it is understandably very descriptive, but nonetheless I would enjoy a little more scientific insights/depth. The future research directions would be the easiest section to insert some creativity, but is very generic; it seems to suggest we need more data from more suitable samples with better descriptions, as could apply anywhere. From a (seismo-)tectonic perspective, I think it would be interesting to investigate how these RSL estimates link with fault slip rates and earthquake activity, and from a geodynamic perspective it would be interesting to understand where the 5e shorelines are in the W half of your study area. According to Sarr et al., 2019, https://doi.org/10.1130/G45629.1) the Sunda Shelf is subsiding, and RSL markers should thus be looked for below present-day sea-level. These are just some suggestions, but in general I think the paper is in a good state to be published, with only some minor remarks for the authors' consideration.*

Response: We thank Dr. Gino de Gelder #1 for the insightful review and helpful suggestions. For the Future research directions section, we tried to address the suggestion of Dr. de Gelder and provided some additional insights on the compiled LIG sea-level indicators in Southeast Asia (Line 464). For this data paper, we did not include a discussion on how these paleo sea-level estimates link with fault slip rates and earthquake activity in the region since this is beyond the scope of this paper and of this journal. The paleo sea level we compiled is still uncorrected for post-depositional land movements (e.g., local or regional tectonic effects, sediment compaction, dynamic topography) or glacial isostatic adjustment (GIA) effects; doing so would entail a more detailed study which is no longer within the scope of our data paper. We encourage future researchers to use the standardized data we compiled in their examination of the effects of post-depositional land movements or GIA in Southeast

Asia. To address this comment, we included an explanation of the scope of the paper in Section 8: Future research directions (Line 466).

Line 477: For this compilation, most of the LIG (MIS 5e) sea-level indicators were found in the eastern portion of Southeast Asia, which is situated in the direct vicinity of major tectonic plates. We did not encounter available publications on LIG sea-level indicators in the western portion of Southeast Asia, which lies inside the subsiding Sundaland block. A tidal notch from Palawan Island in the Philippines, which lies on the eastern boundary of the Sundaland block, reveals MIS5e RSL at 6.8±0.6 m (WALIS RSL ID 1349). The presence of an emergent LIG sea-level indicator within its eastern border as opposed to the lack of evidence of higher-than-present sea level within its western portion (e.g., mainland Southeast Asia, peninsular Malaysia, Sunda shelf) reveals that the Sundaland block might have also experienced varying sea-level histories from site to site. To understand this better, it is worthwhile to re-visit previously studied sites and explore select sites in the Sundaland block (e.g., mainland Southeast Asia) for future work. Documenting emergent LIG sea-level indicators in these locations might help unravel the long-term tectonic behavior of this region.

Line 488: In this paper, we did not include a discussion on how the elevation of paleo RSL proxies links with uplift rates across Southeast Asia, as these are endeavors that go beyond the description of our database. However, we remark that the paleo sea level estimates we report are still uncorrected for post-depositional land movements (e.g., local or regional tectonic effects, sediment compaction, dynamic topography) or glacial isostatic adjustment (GIA) effects. Future studies might be directed towards exploring the magnitude of GIA in this region and using our data to gather the magnitude of other post-depositional movements, comparing them with datasets from structural geology (e.g., fault slip rates and earthquake activity) or even from short-term datasets, such as land motion rates derived from GPS datasets.

*Minor comments:*

*Abstract: I think it's useful to mention as well you have included some MIS 5a and 5c RSL estimates.*

R: We agree with this comment and revised the abstract accordingly.

Line 17: Five data points date to MIS 5a (80 ka), six data points are MIS 5c (100 ka), and the rest are dated to MIS 5e.

*Line 33: 14 what? Publications?*

R: We revised the text accordingly.

Line 35: We screened a total of 14 published studies addressing geological descriptions of LIG sea-level indicators.

*Line 36: change to "Despite our work mostly being aimed"*

R: We accepted this comment and revised the text accordingly.

*Line 37: change to "also inserted"*

R: We adopted this suggestion and revised the text accordingly.

Line 38-40: Despite our work mostly being aimed at compiling MIS 5e data, we also inserted in our database MIS 5a and MIS 5c sites whenever they were not already in the database at the time of compilation (e.g., Thompson and Creveling, 2021).

*Literature overview: I think it's good to mention here that the Sunda Shelf is largely subsiding (Sarr et al. and references therein, see above), which directly explains the absence of RSL markers in a large part of the map.*

R: We adopted this suggestion and included a brief discussion of the Sundaland in the Literature Overview section. Accordingly, the figure is also edited to show the extent of the Sundaland block.

Line 90: While the eastern portion of Southeast Asia is situated at the junction of major tectonic plates, the western portion of the region lies inside the Sundaland block. The core of Sundaland block encompasses mainland Southeast Asia (Cambodia, Laos, Vietnam, Thailand), the Malay Peninsula, Borneo, Sumatra, Java, and the Sunda shelf and has low rate of shallow seismicity (Simons et al., 2007). Recent work, which combined geomorphological observations with numerical simulations of coral reef growth and shallow seismic stratigraphy, suggests that the Sundaland is subsiding with transient dynamic topography the likely cause of subsidence (Sarr et al., 2019 and references therein).

*Line 72: change to "has resulted"*

R: We adopted this suggestion and revised the text accordingly.

Line 77: Oblique convergence between these two plates has resulted in the formation of the Philippine Mobile Belt (PMB), a 400-km-wide deformation zone from Luzon to Mindanao (Philippines), that is bounded by oppositely-dipping subduction zones (the discontinuous Manila- Sulu- Negros-Cotabato Trench system to the west and the East Luzon Trough-Philippine Trench system to the east) (Gervasio, 1966; Rangin et al., 1999).

*Line 159: "reflects sea-level stillstands" sounds vague to me. How about "is proportional to the duration of sea-level stillstands?"*

R: We agree with this comment and revised the text accordingly.

Line 177: The notch depth, which is the horizontal distance between the retreat point and the projected vertical plane from the edge of the roof, is proportional to the duration of sea-level stillstands (Pirazzoli, 1986; Antonioli et al., 2015).

*Line 170: change to "as 20%"*

R: We agree with this comment and revised the text accordingly.

Line 188: If the elevation measurement method was not reported by the original study, the elevation error was calculated as 20% of the reported elevation.

*Line 235: I would change tectonics (very generic) to "active tectonic"*

R: We agree with this comment and revised the text accordingly.

Line 253: One of the earliest works reporting LIG sea-level indicators in Southeast Asia was that of Chappell and Veeh (1978), examining coral reef terraces in East Timor (Atauro Island and north Timor Island) to assess RSL changes within the active tectonic context of the study area.

*Line 450: Not sure if there is a standard format, but to me it makes more sense to address Future research directions (scientific) before the Data availability (technical details)*

R: We agree with this suggestion and revised the organization of these two sections (Line 454-486).

*Line 451: change to "reveals"*

*Line 452: change "done" with "obtained"*

*Line 454: remove "made"*

R: We agree with some of these suggestions and revised the text accordingly.

Line 467: The present standardized database of LIG sea-level indicators in SE Asia compiles available studies on coral reef terraces and tidal notches in the Philippines, Sulawesi, and Sumba-Timor-Alor regions. While reliable

ages were obtained by U-Th-dating of fossil corals, challenges still persist due to preservation potential of corals and the difficulty of finding pristine, diagenetically unaltered and thus dateable samples. The calculation of paleo sea level from coral reef terraces is difficult since vertical uncertainties from measurement techniques are not mentioned in many cases and the indicative meaning of the sea-level indicators is not well documented.

*In the data table: small typo throughout when writing "Chutcharavan adn Dutton"*

R: We agree with this comment and a revised version of the database (reflecting changes about this typo) is uploaded.

Line 605: Maxwell, K., Westphal, H. and Rovere, A.: Database of Last Interglacial (MIS 5e) Sea-level Indicators in Southeast Asia, doi:10.5281/zenodo.5040784, 2021.

---

## Author Response (AR2)

*Professor Matteo Vacchi*
**Editor**
WALIS – the World Atlas of Last Interglacial Shorelines
Earth System Science Data

Dear *Prof. Vacchi*,

Please find enclosed the revised manuscript entitled "**A standardized database of Last interglacial (MIS 5e) sea-level indicators in Southeast Asia**" to be considered for publication in the Special Issue "WALIS – the World Atlas of Last Interglacial Shorelines" under the Earth System Science Data journal.

We thank you for dedicating the time in thoroughly reviewing our manuscript. We revised the manuscript accordingly based on your additional comments and suggestions. For this, we generated a new figure (Figure 3, pages 12-13) to better visualize the variability of MIS 5 shorelines in the region. We hope that with this, the reader will be able to clearly picture the elevations of the MIS 5 RSL indicators alongside the general tectonics of Southeast Asia. Additionally, we have double-checked the dataset carefully and it is coherent with the SI guidelines.

With our best regards,

Kathrine Maxwell*, Hildegard Westphal, Alessio Rovere
*corresponding author